# Nutritional Challenges Among Children Under Five in Limpopo Province, South Africa: Complementary Feeding Practices and Dietary Diversity Deficits

**DOI:** 10.3390/nu17111919

**Published:** 2025-06-03

**Authors:** Tshilidzi Mafhungo, Lindiwe Priscilla Cele, Mmampedi Mathibe, Perpetua Modjadji

**Affiliations:** 1Department of Public Health, School of Health Care Sciences, Sefako Makgatho Health Sciences University, 1 Molotlegi Street, Ga-Rankuwa 0208, South Africa; 2Non-Communicable Diseases Research Unit, South African Medical Research Council, Tygerberg, Cape Town 7505, South Africa; 3Department of Life and Consumer Sciences, College of Agriculture and Environmental Sciences, University of South Africa, Florida Campus, Johannesburg 1709, South Africa

**Keywords:** complementary feeding, dietary diversity, nutritional status, child, rural population, South Africa

## Abstract

**Objective:** The aim of this study was to assess complementary feeding practices and dietary diversity in relation to the nutritional status of children under five attending health facilities in the Thabazimbi sub-district, Limpopo Province. **Methods:** A cross-sectional study was conducted among 409 mother–child pairs. Data on socio-demographics, feeding practices, and anthropometry were collected using validated tools. Nutritional status was assessed using WHO growth standards, and dietary diversity was evaluated using WHO infant and young child feeding (IYCF) indicators and a 24 h dietary recall. Associations were analyzed using prevalence ratios in STATA 18. **Results:** Among 409 children (median age: 18 months, IQR: 12–24), 38% were stunted, 13% were underweight, 5% were thin, and 17% were overweight/obese. Exclusive breastfeeding was reported in 27%, and only 24% met the minimum dietary diversity (DDS ≥ 4). Complementary feeding practices varied significantly by maternal age, with mixed feeding more common among older mothers and younger mothers more likely to receive feeding advice (*p* = 0.001). Stunting was associated with being a boy (PR = 1.27; 95% CI: 1.00–1.61), age > 24 months (PR = 0.33; 95% CI: 0.16–0.65), and DDS ≥ 4 (PR = 0.72; 95% CI: 0.52–0.99). Underweight was more prevalent among boys (PR = 2.40; 95% CI: 1.40–4.11), but less likely in children with DDS ≥ 4 (PR = 0.43; 95% CI: 0.20–0.92) and those from spouse-headed households (PR = 0.33; 95% CI: 0.13–0.87). Thinness was associated with DDS ≥ 4 (PR = 2.70; 95% CI: 1.13–6.45) and age 12–24 months (PR = 2.80; 95% CI: 1.02–7.64). Overweight/obesity was linked to age 12–24 months (PR = 1.94; 95% CI: 1.25–3.03) and household income > ZAR 15,000 (PR = 4.09; 95% CI: 2.33–7.17). **Conclusions:** Complementary feeding and dietary diversity deficits contribute significantly to the dual burden of malnutrition in rural Limpopo, highlighting the need for targeted, context-specific nutrition interventions.

## 1. Introduction

Childhood malnutrition remains a pressing public health concern globally, particularly in low- and middle-income countries (LMICs) where both undernutrition and overnutrition coexist [1,2]. Nutritional status, referred to as the condition of health influenced by nutrient intake and utilization, is a key indicator of child well-being and development [3]. Optimal nutritional status is achieved when children receive adequate amounts of both macro- and micronutrients to support growth, immune function, and cognitive development [4,5]. Nutritional status is commonly assessed using anthropometric measures such as weight-for-age, height/length-for-age, and weight-for-height, which reflect different forms of malnutrition, including, stunting, underweight, wasting (i.e., thinness), and overweight [6]. However, in many parts of sub-Saharan Africa (SSA), children face a dual burden of malnutrition, where chronic undernutrition, including stunting (approximately 32%), underweight (17%), and wasting (6.8%), coexists with rising rates of childhood overweight and obesity, now affecting about 5% of children under five [7,8].

The complex and layered determinants of child malnutrition are emphasized in the Developmental Origins of Health and Disease (DOHaD) framework, namely, that nutritional exposures during critical windows, especially the first 1000 days from conception to age two, can have lasting effects on growth, metabolism, and disease risk [9]. Furthermore, the UNICEF and WHO conceptual models, in addition to the stunting framework, collectively distinguish [7,10,11] between immediate causes (e.g., inadequate dietary intake and illness), underlying factors (e.g., food insecurity, caregiving practices, and healthcare access), and broader structural drivers such as poverty, education, and governance. UNICEF further highlights the role of food systems, caregiving environments, and essential services in shaping maternal and child nutrition [7].

Complementary feeding typically begins around six months of age, when breast milk alone no longer meets an infant’s nutritional needs [12,13]. Timely introduction of solid or semi-solid foods, appropriate meal frequency, and continued breastfeeding up to two years are critical for healthy growth [12,13,14]. However, early diets are often dominated by maize porridge and commercial cereals, with limited inclusion of nutrient-dense foods [13,15]. These dietary patterns are influenced by caregiver knowledge, socio-economic status, and cultural norms [3,14,16,17]. Suboptimal complementary feeding practices among children under five in SSA have been linked to increased risks of stunting, wasting, and micronutrient deficiencies, which can impair cognitive development and increase susceptibility to infections [18,19,20].

Dietary diversity plays a vital role in early childhood nutrition, ensuring intake from a variety of food groups to meet nutrient requirements and prevent deficiencies [21,22,23,24]. The Dietary Diversity Score (DDS), based on a 24 h recall, is a widely used proxy for diet quality [25,26]. WHO recommends a minimum dietary diversity (MDD) of five out of eight food groups for children aged 6–23 months [6]. Higher DDS is consistently linked to better nutritional outcomes, including reduced stunting and underweight [22,23]. Conversely, diets dominated by starchy staples and lacking in fruits, vegetables, and animal-source foods are associated with poor growth and micronutrient deficiencies [21,22,27]. Therefore, adequate dietary diversity is crucial for preventing poor nutritional outcomes, which are prevalent in LMICs [28,29,30].

Despite national and global efforts to improve child nutrition, rural areas in South Africa continue to face significant challenges. In Limpopo Province, poor complementary feeding practices and low dietary diversity are prevalent, with many children introduced to complementary foods too early or too late, and diets are dominated by nutrient-poor staples [14,17,31]. These practices contribute to high rates of stunting, underweight, and overweight, reflecting the coexistence of under- and overnutrition [17,32]. Limpopo, one of the most rural and economically disadvantaged provinces, faces limited access to diverse foods, cultural barriers, and under-resourced health services, and there is a scarcity of data on complementary feeding and dietary diversity, despite several studies that determined nutritional status [14,17,23,33]. Given these gaps, this study hypothesized that suboptimal to complementary feeding practices and low dietary diversity are associated with poor nutritional outcomes among children under five in rural Limpopo. Therefore, the objective of this study was to assess complementary feeding practices and dietary diversity in relation to the nutritional status of children under five attending health facilities in the Thabazimbi sub-district, Limpopo Province, South Africa. The findings aim to inform context-specific public health strategies that address early-life nutritional challenges and support optimal growth and development.

## 2. Materials and Methods

### 2.1. Study Design

This study employed a cross-sectional analytical design to examine the association between complementary feeding practices, dietary diversity, and the nutritional status of children under five attending primary healthcare facilities for routine child health services, including immunization and growth monitoring. The study was conducted between March and August 2023.

### 2.2. Study Setting

The study was conducted in the Thabazimbi sub-district, situated within the Waterberg District of Limpopo Province, South Africa. Thabazimbi is one of five sub-districts in the district and is characterized by a diverse geographic and socio-economic landscape, encompassing rural villages, semi-rural settlements, and mining communities. The sub-district has an estimated population of approximately 105,000 residents, with Setswana being the predominant language spoken. Geographically, it lies approximately 235 km southwest of Polokwane, the provincial capital, and about two hours by road from Pretoria. Thabazimbi is notable for its iron mining industry and shares a border with Botswana, contributing to its demographic and economic complexity. The sub-district comprises 10 public health facilities strategically distributed across farming areas, informal settlements, mining zones, and peri-urban communities, where the study was conducted. These facilities provide a comprehensive range of primary healthcare services, including maternal and child health, nutrition counseling, immunization, chronic disease management, and emergency care. The setting reflects the broader challenges of rural health service delivery in South Africa, including limited access to diverse foods, under-resourced health infrastructure, and socio-economic disparities that influence child nutrition and feeding practices.

### 2.3. Study Population

The study targeted biological mothers of children under five years of age who attended selected public health facilities in the Thabazimbi sub-district for routine growth monitoring and immunization services. These visits were part of standard child healthcare, and the mothers themselves were not seeking care for illness. These mothers were considered the most appropriate respondents, as mothers are typically the primary caregivers and are directly responsible for infant and young-child feeding practices, including decisions related to complementary feeding and dietary choices. The study population was drawn from an estimated 10,000 to 12,000 children under five years of age who regularly access child health services across the sub-district’s ten primary healthcare facilities for routine child health services, including immunization and growth monitoring. Inclusion criteria required that participants be the biological mother of a child aged 0–59 months, be present at the facility on the day of data collection and provide informed consent. Mothers who were not the primary caregivers or whose children had known chronic illnesses or congenital conditions affecting growth or feeding were excluded. The sample reflected the population most directly involved in early-life nutrition and caregiving within the local context.

### 2.4. Sample Size and Sampling Procedure

The sample size for this study was calculated using the RAO software (http://www.raosoft.com/samplesize.html, accessed on 29 May 2021) sample size calculator (Raosoft Inc., Seattle, WA, USA, 2004), based on an estimated population of 10,000 to 12,000 children under five years of age attending public health facilities in the Thabazimbi sub-district for routine child health services, including immunization and growth monitoring. The calculation assumed a 5% margin of error, a 95% confidence level, and a 50% response distribution, yielding a minimum required sample size of 373 participants. To account for potential non-responses and incomplete questionnaires, a 10% contingency (*n* = 37) was added, resulting in a final target sample size of 410 mother–child pairs.

A multistage sampling strategy was employed to ensure representativeness across the sub-district. In the first stage, health facilities were selected using stratified random sampling, considering geographic distribution, patient volume, and accessibility. A list of all public health facilities in the sub-district was stratified by location (rural, peri-urban, and mining communities), and facilities were randomly selected from each stratum using a random number generator to ensure geographic and service-level diversity. In the second stage, systematic random sampling was used to recruit eligible participants within each facility. Upon arrival at each clinic, a sampling interval was determined based on the average daily number of eligible mothers attending child health services. Every third biological mother in the clinic waiting area was approached after completing her consultation and invited to participate in the study. The starting point was randomly selected each day by drawing a number between 1 and 3.

Recruitment was conducted across three clinics: 222 mother–child pairs were initially approached at Clinic 1, 128 at Clinic 2, and 243 at Clinic 3. Following exclusions due to ineligibility or refusal to participate (82, 28, and 73 pairs, respectively), the final sample comprised 140 participants from Clinic 1, 100 from Clinic 2, and 170 from Clinic 3, totaling 410 mother–child pairs. As explained above, the inclusion criteria required participants to be biological mothers aged 18 years or older, with children aged 0–59 months, attending the clinic for child health services for routine child health services, including immunization and growth monitoring, and willing to provide informed consent. It is important to clarify that the mothers themselves were not ill; they were attending clinics to access routine child health services such as immunization, growth monitoring, and nutrition counseling. Exclusion criteria included mothers under 18 years of age, those medically unfit or unable to breastfeed due to illness, and those whose children had an acute illness or congenital conditions affecting feeding or growth. A representative and unbiased sample of the target population was ensured, facilitating the collection of valid and generalizable data. The recruitment process is illustrated in Figure 1.

### 2.5. Data Collection and Tools

The research adhered to the STROBE (Strengthening the Reporting of Observational Studies in Epidemiology) guidelines to ensure methodological transparency, rigor, and completeness in the presentation of the methods and findings [34].

To ensure the reliability and validity of the data collection process, several quality control measures were implemented. Data were collected using a structured questionnaire adapted from validated tools in maternal and child nutrition research, informed by the DOHaD framework and the conceptual models of UNICEF and WHO. These frameworks collectively emphasize the importance of early-life nutritional exposures and the multi-layered determinants of child malnutrition. Content and face validity were established through expert review to ensure that the tool accurately captured relevant constructs. To enhance cultural and linguistic appropriateness, the questionnaire was translated from English into Setswana by a bilingual translator. A pilot study involving 5% of the target sample [35] was conducted at a non-participating facility to assess clarity, feasibility, and cultural relevance. Minor revisions were made based on pilot feedback.

The study was conducted in public health facilities (clinics), where data were collected through direct interviews using the structured questionnaire. The interviews were conducted by the principal researcher and two trained research assistants. Research assistants fluent in both Setswana and English received one week of training on standardized data collection procedures, including anthropometric measurements and interview techniques. During the pilot, their performance was assessed to ensure consistency and accuracy in administering the questionnaire and taking measurements. All anthropometric tools were calibrated, and measurements were taken twice to minimize error. The final version of the tool was administered in Setswana by a trained research assistant.

#### 2.5.1. Socio-Demographics of Study Participants

Socio-demographic data were collected using a structured questionnaire adapted from previously validated instruments used in maternal and child nutrition research [17,31,36,37]. Informed by the conceptual models of UNICEF and WHO, the questionnaire was designed to capture a comprehensive profile of the participants and contextual factors influencing child nutrition. Key domains included maternal characteristics (e.g., age, marital status, education level, and employment status), household information (e.g., income, family size, housing type, and access to electricity and water), obstetric history, and child-specific variables (e.g., age and sex). These variables were selected to reflect both immediate and underlying determinants of malnutrition, consistent with the study’s conceptual framework.

#### 2.5.2. Complementary Feeding Practices and Dietary Diversity

Complementary feeding practices were assessed using a structured questionnaire adapted from WHO guidelines on Infant and Young Child Feeding (IYCF) [38,39]. Mothers provided a 24 h dietary recall of all foods and beverages consumed by their child, enabling the evaluation of key IYCF indicators, including the timing of complementary food introduction, feeding frequency, food types, and continued breastfeeding. These indicators were assessed in accordance with WHO recommendations, which emphasize timely, adequate, and safe introduction of nutrient-dense foods beginning at six months of age. According to WHO recommendations, complementary feeding should begin at six months of age, as breast milk alone is no longer sufficient to meet the nutritional needs of growing infants. Complementary foods should be introduced in a timely, adequate, safe, and appropriate manner. This includes providing nutrient-dense foods in age-appropriate textures and frequencies, 2 to 3 times per day for infants aged 6 to 8 months, increasing to 3 to 4 times daily for children aged 9 to 23 months, with 1 to 2 additional nutritious snacks per day for those aged 12 to 24 months [40].

Dietary diversity was measured using the WHO-recommended 24 h recall method, and the Dietary Diversity Score (DDS) was calculated based on consumption from seven food groups, following FAO guidelines [41]. The MDD indicator was defined as the proportion of children who received food from at least four of the seven food groups. These food groups included the following: (1) grains, roots, and tubers; (2) legumes and nuts; (3) dairy products; (4) flesh foods; (5) eggs; (6) vitamin A-rich fruits and vegetables; and (7) other fruits and vegetables. A DDS of ≤4 was classified as low dietary diversity, a threshold commonly used in similar studies to indicate poor dietary quality [42]. The minimum meal frequency (MMF) was also assessed, defined as the number of times children received solid, semi-solid, or soft foods (including milk feeds for non-breastfed children) appropriate to their age and breastfeeding status. The minimum acceptable diet (MAD) was derived by combining MDD and MMF indicators.

#### 2.5.3. Anthropometric Measurements and Nutritional Indicators of Children

Anthropometric measurements of children were conducted by the research assistants following standardized procedures [43]. Weight was measured using a calibrated Seca 354 digital baby scale (Seca GmbH & Co. KG, Hamburg, Germany) for all children, with the child wearing minimal clothing and no shoes. Measurements were recorded to the nearest 0.1 kg. For children under two years of age, recumbent length was measured using a Seca 210 infantometer (Seca GmbH & Co. KG, Hamburg, Germany), with the child lying flat on their back, legs fully extended, and head positioned against the fixed headboard. For children aged two years and older, standing height was measured using a portable Seca stadiometer (Seca GmbH & Co. KG, Hamburg, Germany), with the child standing upright, barefoot, heels together, and head in the Frankfort horizontal plane. Height and length were recorded to the nearest 0.1 cm. All anthropometric measurements were taken twice by trained fieldworkers, and the average of the two readings was used in the analysis. If the two measurements differed by more than 0.1 kg for weight or 0.5 cm for length/height, a third measurement was taken, and the two closest values were averaged. The collected data were entered into the WHO Anthro software version 3.2.2 to calculate sex-specific Z-scores for weight-for-age (WAZ), height/length-for-age (HAZ/LAZ), and BMI-for-age (BAZ). Nutritional status was classified according to WHO Child Growth Standards as follows: stunting was defined as HAZ/LAZ less than −2 standard deviations (SDs), underweight as WAZ less than −2 SDs, wasting or thinness as BAZ less than −2 SDs, overweight as BAZ greater than +2 SDs and less than or equal to +3 SDs, and obesity as BAZ greater than +3 SDs [39].

#### 2.5.4. Anthropometric Measurements of Mothers

Anthropometric measurements of the mothers were conducted by the principal researcher using standardized procedures as recommended by the World Health Organization (WHO) [44] to ensure accuracy, consistency, and comparability. All measurements were taken in a private setting to ensure the participants’ comfort and confidentiality. Body weight was measured using a calibrated Beurer GS 203 digital scale (Beurer GmbH, Ulm, Germany). Participants were asked to remove shoes, heavy clothing, and accessories before stepping onto the scale. They stood upright in the center of the scale platform, with arms relaxed at their sides and looking straight ahead. The scale was placed on a firm, level surface, and weight was recorded to the nearest 0.1 kg. The scale was checked for calibration daily using a standard weight. Height was measured using a portable Seca stadiometer (Germany). Participants were instructed to stand barefoot with their heels together, legs straight, arms at their sides, shoulders relaxed, and back against the stadiometer’s vertical board. The head was positioned in the Frankfort horizontal plane (a line from the lower margin of the orbit to the upper margin of the ear canal). The movable headpiece was gently lowered to rest on the crown of the head, and height was recorded to the nearest 0.1 cm. The stadiometer was placed on a flat, stable surface and checked for vertical alignment before each session. Each anthropometric measurement (weight and height) was taken twice. If the two readings differed by more than 0.1 kg for weight or 0.5 cm for height, a third measurement was taken, and the two closest values were averaged for analysis. Body mass index (BMI) was calculated using the standard formula: weight in kilograms divided by height in meters squared (kg/m^2^). Based on WHO classification criteria, BMI was categorized as follows: underweight (<18.5 kg/m^2^), normal weight (18.5–24.9 kg/m^2^), overweight (25.0–29.9 kg/m^2^), and obese (≥30.0 kg/m^2^) [45].

### 2.6. Statistical Analysis

Data were initially captured in Microsoft Excel and analyzed using STATA 18 (StataCorp, College Station, TX, USA). After reviewing all questionnaires for completeness, one was excluded due to missing primary outcome data, resulting in a final analytical sample of 409 mother–child pairs. Anthropometric indices (HAZ/LAZ, WAZ, and BAZ) were calculated using WHO Anthro software version 3.22, based on age- and sex-specific Z-scores. The distribution of continuous variables was assessed using the Shapiro–Francia test for normality, and non-parametric tests (Mann–Whitney U and Kruskal–Wallis) were applied where appropriate. Categorical variables were summarized as frequencies and percentages, with group comparisons conducted using Pearson’s chi-square test or Fisher’s exact test when expected cell counts were below five. Socio-economic and feeding practice variables were compared across child age groups, and nutritional indicators were examined by sex and age group. To explore associations between nutritional outcomes and explanatory variables, univariate and multivariate Poisson regression models with robust standard errors were used to estimate crude (PR) and adjusted prevalence ratios (aPR) with 95% confidence intervals (CI). Variables with *p*-values ≤ 0.25 in univariate analysis were included in the multivariate models using a stepwise backward elimination approach. Potential confounders at the child, maternal, and household levels were adjusted for in the final models to improve the accuracy of the estimates. Results are presented as medians with interquartile ranges (IQRs) for continuous variables, frequencies and percentages for categorical variables, and PR and aPR with 95% CIs for regression models. Statistical significance was set at *p* < 0.05.

## 3. Results

### 3.1. Characteristics of Mothers

The study analyzed 409 mother–child pairs in Thabazimbi, categorized by maternal age: <25 years (G1), 25–34 years (G2), and ≥35 years (G3). Significant differences across the three age groups (P^0^) were observed in dwelling place (*p* = 0.023), refrigerator use (*p* = 0.050), and obstetric complications (*p* = 0.003). BMI distribution did not differ significantly (*p* = 0.090), though overweight and obesity were slightly more prevalent in G1 (28%) compared to G3 (25%). Most mothers were single (81%), had secondary education (69%), and were unemployed (93%), with no statistically significant variation by age group. Older mothers (G3) were more likely to receive child grants (94% vs. 79% in G1, *p* = 0.006) and reside in brick houses (55% vs. 38%, *p* = 0.023). Obstetric complications were more common among younger mothers (G1) compared to older groups (*p* = 0.003). Other reproductive indicators, such as parity, pregnancy term, and maternal age at first birth, as well as access to electricity, water, and refrigerator use, showed no significant differences in pairwise comparisons. Household income and size were generally similar, although a trend toward higher income among older mothers was observed. These findings are detailed in Table 1.

### 3.2. Characteristics of Children

The study included 409 children attending clinics in the Thabazimbi sub-district, Limpopo Province, with a median (interquartile range; IQR) age of 18 months (12; 24), ranging from 12 to 60 months. Anthropometric assessments showed a median weight of 9.9 kg (8.2; 11.5), with values ranging from 6 to 21 kg, and a median length of 76.1 cm (69; 80), ranging from 60 to 111 cm. Z-score distributions indicated a wide range, with LAZ/HAZ scores from −6.2 to 6, WAZ scores from −4.14 to 4.22, and BAZ scores from −4.49 to 6.31 (Table 2).

### 3.3. Comparison of the Nutritional Status of Children by Sex and Age

Among 409 children (51% girls, 49% boys), stunting (38%) was significantly more common in boys (45%) than girls (32%) (*p* = 0.034). Underweight affected 13% overall and was also higher in boys (18%) than girls (8%) (*p* = 0.004). No significant sex differences were found for tallness, thinness, or overweight/obesity. By age, 69% were ≤12 months, 17% were 13–24 months, and 13% were >24 months. Stunting was significantly more prevalent in younger age groups (41% in ≤12 months, 47% in 13–24 months) compared to older children (13%) (*p* = 0.001). Overweight/obesity was highest in the 13–24-month group (20%) (*p* = 0.038), while other indicators showed no significant age-related differences (Table 3).

### 3.4. Infant and Complementary Feeding Practices

Among the 409 mother–child pairs, breastfeeding and complementary feeding (CF) practices varied by maternal age. While 98% of children had been breastfed, only 27% were exclusively breastfed, and continued breastfeeding up to one year was not practiced. Complementary feeding was often introduced before six months, with 20% of mothers reporting no advice received. Significant differences across age groups were observed in three areas: source of breastfeeding influence (*p* ≤ 0.0001), with older mothers more influenced by healthcare workers (HCW); mixed feeding practices (*p* = 0.001), more common among older mothers; and receipt of complementary feeding advice (*p* = 0.001), with younger mothers more likely to report receiving guidance. These findings suggest age-related disparities in feeding practices and access to health information (Table 4).

### 3.5. Dietary Diversity and Food Group Consumption Among Children

Dietary diversity was assessed based on the consumption of seven recommended food groups within the preceding 24 h. DDS ranged from 2 to 7, calculated by summing the number of distinct food groups consumed. MDD was defined as the consumption of at least four food groups: met MDD ≥ 4 food groups and unmet MDD < 4 food groups. Findings from the current study revealed that 24% of children did not meet the MDD threshold (Figure 2). The mean DDS was 3.06 (±0.49).

The food groups most consumed by children were grains, roots, and tubers (98%), followed by vitamin A-rich dairy products (67%), legumes and beans (56%), and other fruits and vegetables (51%). In contrast, the consumption of flesh foods, eggs, and vitamin A-rich fruits and vegetables was below 50% (Figure 3).

### 3.6. Multivariate Analysis of Factors Associated with Child Nutritional Status

Table 5 presents prevalence ratios for factors associated with stunting, underweight, thinness, and overweight/obesity among children under five. Variables with a *p*-value ≤ 0.25 in bivariate analysis were included in the final model. Stunting was significantly associated with being a boy (aPR = 1.27; 95% CI: 1.00–1.61), being older than 24 months (aPR = 0.33; 95% CI: 0.16–0.65), and achieving a dietary diversity score (DDS) of four or more (aPR = 0.72; 95% CI: 0.52–0.99). Underweight was more likely among boys (aPR = 2.40; 95% CI: 1.40–4.11), but less likely in children with higher DDS (aPR = 0.43; 95% CI: 0.20–0.92) and those living in spouse-headed households (aPR = 0.33; 95% CI: 0.13–0.87). Thinness was associated with DDS ≥ 4 (aPR = 2.70; 95% CI: 1.13–6.45) and being aged 12–24 months (aPR = 2.80; 95% CI: 1.02–7.64). Overweight/obesity was significantly associated with being aged 12–24 months (aPR = 1.94; 95% CI: 1.25–3.03) and living in households with monthly incomes above ZAR 15,000 (aPR = 4.09; 95% CI: 2.33–7.17).

## 4. Discussion

This study investigated complementary feeding practices and dietary diversity in relation to the nutritional status of children under five in the Thabazimbi sub-district, Limpopo Province. The findings show persistent nutritional challenges, characterized by suboptimal IYCF practices, limited dietary diversity, and a dual burden of malnutrition, emphasizing the need for targeted, context-specific public health interventions.

The socio-demographic profile of participating mothers reflects broader structural inequalities. The majority were young, single, unemployed, and reliant on social grants, conditions that have been consistently linked to food insecurity and poor child nutrition outcomes [32,37]. Although social protection mechanisms such as child support grants provide essential income, they remain insufficient to ensure dietary adequacy, particularly in rural settings with limited access to diverse foods and basic services [10,46,47]. Significant differences in types of dwelling houses, refrigerator access, and obstetric complications across maternal age groups in this study further emphasize disparities in household infrastructure and maternal health, which may influence feeding practices and child outcomes, as reported earlier [48,49,50].

Anthropometric assessments of children in this study showed a high prevalence of poor nutritional status: stunting (38%), followed by underweight (13%), thinness (5%), and overweight/obesity (17%). These findings are consistent with national and regional data [51,52] and reflect the coexistence of chronic undernutrition and emerging overnutrition in low-resource settings [32]. Although underweight and wasting were less common, they remain critical indicators of both acute and chronic food insecurity, often associated with poor maternal education, inadequate sanitation, and suboptimal feeding practices [53]. The presence of overweight/obesity among children under five, particularly those aged 12–24 months and from higher-income households, aligns with South Africa’s ongoing nutrition transition, marked by increased consumption of energy-dense, nutrient-poor foods [52,54,55]. This dual burden of malnutrition, including the individual-level coexistence of stunting and being overweight, has been associated with increased risk of non-communicable diseases later in life [56,57]. The frameworks applied in this study, informed variable selection, and analysis, helped to clarify the nutritional challenges in this rural context. These findings emphasize the urgent need for integrated nutrition strategies that address both under- and overnutrition, with a focus on improving dietary quality, maternal education, and household food security.

Feeding practices were suboptimal across all age groups in this study. Although breastfeeding initiation was relatively high, exclusive breastfeeding was low (27%), consistent with national trends, despite ongoing public health efforts [58,59]. While initiatives like Mom-Connect aim to promote exclusive breastfeeding [60], their impact is undermined by barriers such as widespread formula marketing, socio-cultural norms, medical conditions, and economic pressures [16,58,59,61,62,63]. Furthermore, complementary feeding was frequently introduced before six months, often driven by perceptions of insufficient breast milk, infant distress, or family influence, factors previously reported in similar contexts [16,17,58,64] as well, highlighting persistent gaps in maternal education and support [13]. Notably, significant differences were observed in the source of breastfeeding influence, with older mothers more likely to be guided by healthcare workers, while younger mothers relied more on family or self [58,65]. Mixed feeding was significantly more prevalent among older mothers, and receipt of complementary feeding advice was more common among younger mothers, suggesting age-related disparities in access to and utilization of health information [66,67]

Dietary diversity is a critical indicator of diet quality and nutritional adequacy in early childhood [22,23,24]. Dietary diversity was limited, with a mean DDS of 3.06, and 24% of children did not meet the minimum dietary diversity (MDD) threshold. Diets were dominated by starchy staples, while consumption of nutrient-dense foods, such as flesh foods, eggs, and vitamin A-rich fruits and vegetables, remained low. Contrary to WHO recommendations, many children are introduced to carbohydrate-rich but nutrient-poor foods, contributing to poor dietary diversity and stunting [68,69]. These patterns are consistent with findings from other sub-Saharan African countries and reflect the influence of food insecurity and limited maternal nutrition knowledge [21,22,70]. Low DDS, a known predictor of micronutrient deficiencies and poor growth outcomes, has been similarly reported in Kenya, Ethiopia, Zambia, Botswana, and Tanzania [70,71,72,73,74].

Multivariate analysis identified dietary diversity as a significant predictor of child nutritional status. Children with a DDS ≥ 4 were 28% less likely to be stunted and 57% less likely to be underweight, highlighting the protective role of dietary diversity against undernutrition [22,73]. Male children had a higher prevalence of both stunting and underweight, while children older than 24 months were less likely to be stunted, suggesting age and sex as important biological determinants [75]. Thinness was more prevalent among children aged 12–24 months and those with higher DDS, possibly reflecting transitional feeding challenges during weaning. Overweight/obesity was significantly more prevalent among children aged 12–24 months and those from higher-income households, indicating early exposure to energy-dense diets and the influence of socio-economic status on dietary patterns [76,77].

This study has several limitations. As a cross-sectional design, it captures associations at a single point in time and cannot establish causality. Reliance on mother-reported data, even when collected through interviews, introduces potential recall and social desirability bias, although food models were used to enhance accuracy. The use of self-reported data may also contribute to measurement error. The study did not quantify actual nutrient intake but used DDS as a proxy for dietary quality. Additionally, several potential confounding variables were not available for inclusion in the multivariate analysis, which may limit the robustness of the findings. The absence of biochemical or clinical assessments further restricts the depth of nutritional evaluation. Also, the findings are specific to a rural setting and may not be generalizable to other populations. Seasonal variation in food availability, which can influence dietary patterns, was also not accounted for. Future research should consider longitudinal designs, incorporate biochemical assessments, and explore regional and seasonal variations for a more comprehensive understanding. Despite these limitations, the study offers valuable insights into complementary feeding and dietary diversity in relation to child nutrition in rural Limpopo. The use of validated tools and a substantial sample size enhances the reliability of the findings.

## 5. Conclusions

This study assessed complementary feeding practices and dietary diversity in relation to the nutritional status of children under five attending health facilities in the Thabazimbi sub-district Limpopo Province. The findings show a high prevalence of both undernutrition (stunting and underweight) and overnutrition (overweight/obesity), highlighting the dual burden of malnutrition in this rural setting. Inadequate complementary feeding practices and limited dietary diversity were common, with dietary diversity emerging as a significant protective factor against stunting and underweight. These results emphasize the importance of promoting diverse and age-appropriate complementary feeding to improve child nutritional outcomes. Targeted, context-specific interventions that support caregiver education, improve access to nutrient-rich foods, and address underlying socio-economic challenges are essential to advancing child health and achieving national and global nutrition targets in underserved communities.

## Figures and Tables

**Figure 1 nutrients-17-01919-f001:**
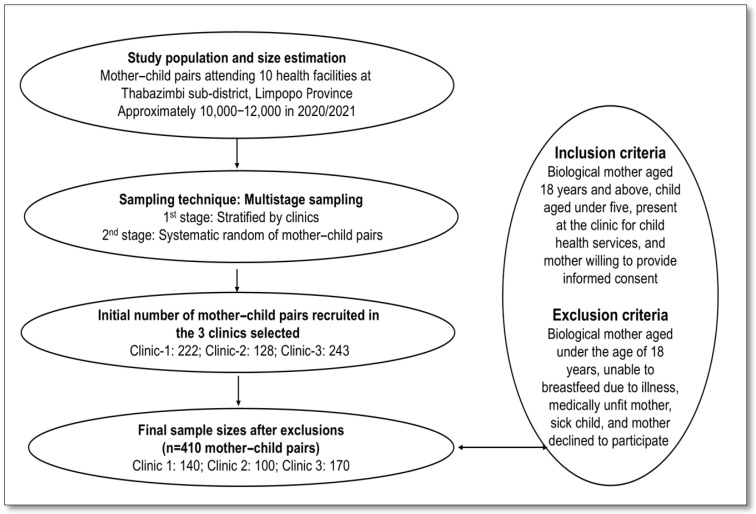
The recruitment process is depicted in the flow chart.

**Figure 2 nutrients-17-01919-f002:**
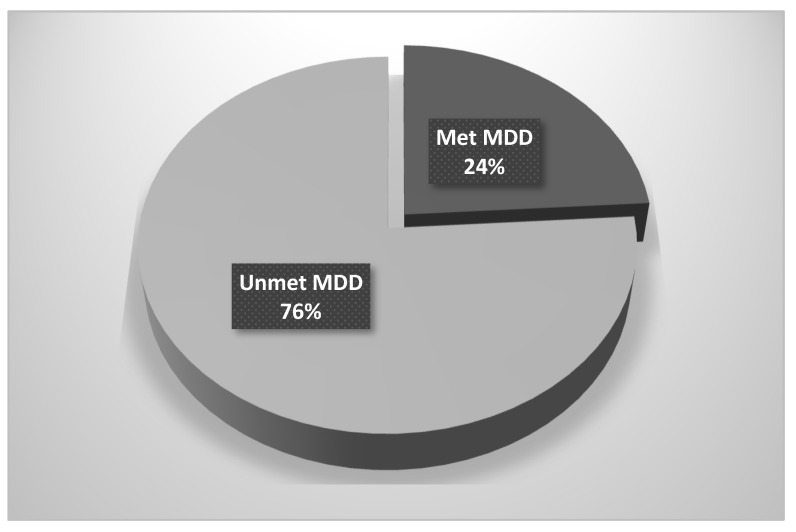
Minimum dietary diversity among children. (MDD stands for minimum dietary diversity.)

**Figure 3 nutrients-17-01919-f003:**
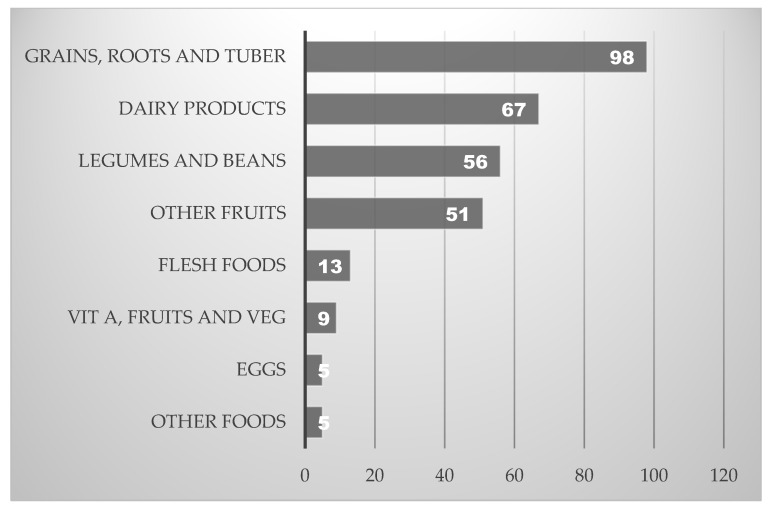
Proportion of each food group consumed in 24 h recall by children.

**Table 1 nutrients-17-01919-t001:** Comparison of characteristics of mothers by age groups.

Variables		All	<25 Years (G1)	25–34 Years (G2)	≥35 Years (G3)	*p*
		*n* (%)	(*n* = 282); *n* (%)	(*n* = 71); *n* (%)	(*n* = 56); *n* (%)	
BMI (Kg/m^2^)	<18.5 (underweight)	10 (2)	7 (3)	1 (1)	2 (4)	P^0^ = 0.090
>18.5–24.99 (normal)		289 (71)	194 (69)	55 (78)	40 (71)	P^1^ = 0.130
25–29.9 (overweight)		87 (21)	66 (23)	14 (20)	7 (13)	P^2^ = 0.901
≥30 (obesity)		23 (6)	15 (5)	1 (1)	7 (12)	P^3^ = 0.319
Marital status	Single	333 (81)	226 (80)	61 (86)	46 (82)	P^1^ = 0.529; P^1^ ≤ 0.0001 *
Ever married		76 (19)	56 (20)	10 (14)	10 (18)	P^2^ ≤ 0.0001 *; P^3^ ≤ 0.0001 *
Education	No school/primary	23 (6)	19 (7)	4 (6)	0 (0)	
Secondary/grade 12		284 (69)	231 (82)	64 (90)	52 (93)	P^0^ = 0.072; P^1^ = 0.130
Post-grade 12		39 (10)	32 (11)	3 (4)	4 (7)	P^2^ = 0.705; P^3^ = 0.107
Employed	No	382 (93)	265 (94)	64 (90)	53 (95)	P^0^ = 0.500; P^1^ = 0.252
Yes		27 (7)	17 (6)	7 (10)	3 (5)	P^2^ = 0.846; P^3^ = 0.352
Child grant	No	69 (17)	58 (21)	8 (11)	3 (5)	P^0^ = 0.006 *; P^1^ = 0.073
Yes		340 (83)	224 (79)	63 (89)	53 (94)	P^2^ = 0.007 *; P^3^ = 0.242
Household head	Self-headed	27 (7)	22 (8)	4 (6)	1 (2)	
Husband/partner		183 (45)	121 (43)	32 (45)	30 (54)	P^0^ = 0.432; P^1^ = 0.872
Parents/relatives		199 (49)	139 (49)	35 (49)	25 (45)	P^2^ = 0.870; P^3^ = 0.790
Household size	<5	182 (44)	130 (46)	24 (34)	28 (50)	P^0^ = 0.118; P^1^ = 0.062; P^2^ = 0.594; P^3^ = 0.067
≥5		227 (56)	152 (54)	47 (66)	28 (50)	
Household income	<ZAR 5000	140 (34)	96 (34)	28 (39)	17 (30)	P^0^ = 0.330
ZAR 5000–ZAR 10,000		187 (46)	129 (46)	31 (44)	29 (52)	P^1^ = 0.368
ZAR 10,001–ZAR 15,000		57 (14)	44 (15)	9 (13)	4 (7)	P^2^ = 0.728
>ZAR 15,000		25 (6)	13 (5)	3 (4)	6 (11)	P^3^ = 0.067
Dwelling place	Non-brick	236 (58)	175 (62)	36 (51)	25 (45)	P^0^ = 0.023 *; P^1^ = 0.082 P^2^ = 0.723; P^3^ = 0.336
Brick house		173 (42)	107 (38)	35 (49)	31 (55)	
Access to electricity	No	9 (2)	7 (3)	2 (3)	0 (0)	P^0^ = 0.668; P^1^ = 0.873 P^2^ = 0.234; P^3^ = 0.207
Yes		400 (98)	275 (97	69 (97)	56 (100)	
Refrigerator use	No	28 (7)	22 (8)	6 (8)	0 (0)	P^0^ = 0.050 *; P^1^ = 0.857 P^2^ = 0.234; P^3^ = 0.207
Yes		381 (93)	260 (92)	65 (92)	56 (100)	
Access to water	No	301 (73)	74 (26)	23 (32)	11 (20)	P^0^ = 0.268; P^1^ = 0.082 P^2^ = 0.723; P^3^ = 0.336
Yes		108 (27)	208 (74)	48 (68)	45 (80)	
Mother’s age at 1st birth	<30	266 (65)	181 (64)	45 (63)	40 (71)	P^0^ = 0.554; P^1^ = 0.900 P^2^ = 0.299; P^3^ = 0.340
≥30		143 (35)	101 (36)	26 (37)	16 (29)	
Parity	1–2	258 (63)	184 (65)	43 (61)	31 (55)	P^0^ = 0.333; P^1^ = 0.462 P^2^ = 0.161; P^3^ = 0.556
≥3		151 (37)	98 (35)	28 (39)	25 (45)	
Pregnancy full term	No	14 (3)	13 (5)	0 (0)	1 (2)	P^0^ = 0.106; P^1^ = 0.066 P^2^ = 0.333; P^3^ = 0.260
Yes		395 (97)	269 (95)	71 (100)	55 (98)	
Obstetric complications	No	337 (82)	223 (79)	60 (85)	54 (96)	P^0^ = 0.003 *; P^1^ = 0.301 P^2^ = 0.002 *; P^3^ = 0.336
Yes		72 (18)	59 (21)	11 (915)	2 (4)	

BMI stands for body mass index; *n* stands for number of participants; ZAR is the currency code for the South African Rand; *p* stands for probability level; G1, G2, and G3 stand for Group 1, Group 2, and Group 3, respectively. P^0^ *p*-value for comparison among the 3 groups; P^1^ between G1 and G2; P^2^ between G1 and G3; P^3^ between G2 and G3; * indicates significant difference.

**Table 2 nutrients-17-01919-t002:** Descriptive statistics of the anthropometric and nutritional indicators of children.

Variables	Median	IQR	Minimum	Maximum
Age (months)	12	12;24	12	60
Weight (kg)	9.9	8.2; 11.5	6	21
Length (cm)	76.1	69; 80	60	111
LAZ/HAZ	−1.37	−2.72; 0.02	−6.2	6
WAZ	−0.46	−1.26; −0.33	−4.14	4.22
BAZ	0.51	−0.42; 1.44	−4.49	−6.31

IQR for interquartile range; LAZ for length-for-age; HAZ for height-forage; WAZ for weight-for-age; and BAZ for BMI-for-age z-scores.

**Table 3 nutrients-17-01919-t003:** Comparison of the nutritional status of children by sex and age.

Variables	All (*n* = 409)	Boys (*n* = 202)	Girls (*n* = 207)	*p*
LAZ/HAZ				0.034 *
Normal Stunted Tallness	236 (58) 157 (38) 16 (4)	106 (52) 90 (45) 6 (3)	130 (63) 67 (32) 10 (5)	0.011 * 0.446
WAZ				0.004 *
Normal Underweight Growth problem	333 (81) 53 (13) 23 (6)	152 (75) 37 (18) 13 (7)	181 (87) 16 (8) 10 (5)	0.001 * 0.481
BAZ				0.506
Normal Thinness Overweight risk Overweight/obesity	247 (60) 19 (5) 79 (19) 64 (17)	116 (57) 11 (5) 39 (19) 36 (18)	131 (63) 8 (4) 40 (19) 28 (14)	0.489 0.997 0.232
**Variables**	**≤12 Months** **(*n* = 282)**	**13–24 Months** **(*n* = 71)**	**>24 Months** **(*n* = 56)**	** *p* **
LAZ/HAZ				0.001 *
Normal Stunted Tallness	155 (55) 117 (41) 10 (4)	35 (49) 33 (47) 3 (4)	46 (82) 7 (13) 3 (5)	≤0.0001 * 0.677
WAZ				0.233
Normal Underweight Growth problem	222 (79) 40 (18) 20 (7)	61 (86) 8 (11) 2 (3)	50 (89) 5 (9) 1 (2)	0.153 0.506
BAZ				0.061
Normal Thinness Overweight risk Overweight/obesity	178 (63) 9 (3) 55 (20) 40 (14)	34 (48) 6 (8) 13 (18) 18 (25)	35 (63) 4 (7) 11 (20) 6 (11)	0.096 0.972 0.038 *

*n* stands for number of participants; * indicates significant difference; LAZ for length-for-age; HAZ for height-for age; WAZ for weight-for-age; and BAZ for BMI-for-age z-score.

**Table 4 nutrients-17-01919-t004:** Complementary feeding and breastfeeding among children.

Variables	All *n* (%)	<25 Years (G1) *n* (%)	25–34 Years (G2) *n* (%)	≥35 Years (G3) *n* (%)	*p*
Advice on CF No Yes	80 (20) 329 (80)	21 (18) 93 (82)	49 (12) 183 (79)	10 (16) 53 (84)	P^0^ = 0.607; P^1^ = 0.167 P^2^ = 0.001 *; P^3^ = 0.149
Introduction of CF (did not exclusively breastfed) <6 months ≥6 months	194 (47) 213 (52)	130 (46) 150 (54)	34 (48) 37 (52)	30 (54) 26 (46)	P^0^ = 0.620; P^1^ = 0.826 P^2^ = 0.329; P^3^ = 0.526
Delayed CF No Yes	403 (98) 6 (2)	277 (98) 5 (2)	70 (99) 1 (1)	56 (100) 0(0)	P^0^ = 0.836; P^1^ = 0.832 P^2^ = 0.316; P^3^ = 0.375
Ever breastfed No Yes	6 (2) 403 (98)	5 (2) 277 (98)	1 (1) 70 (99)	0 (0) 56 (100)	P^0^ = 0.836; P^1^ = 0.832 P^2^ = 0.316; P^3^ = 0.375
Source of influence to breastfeed Self HCW Family	72 (18) 266 (66) 64 (16)	63 (23) 165 (60) 48 (17)	8 (11) 52 (74) 10 (14)	1 (2) 49 (89) 6 (11)	P^0^ ≤ 0.0001 *; P^1^ = 0.292 P^2^ = 0.089; P^3^ = 0.480
Initiation of breastfeeding (colostrum) <1 h 2–3 h ≥4 h After 24 h	287 (71) 79 (19) 28 (7) 12 (3)	191 (68) 55 (20) 22 (8) 11 (4)	50 (70) 16 (23) 5 (7) 0 (0)	46 (82) 79 (19) 1 (2) 1 92)	P^0^ = 0.227; P^1^ = 0.292 P^2^ = 0.089; P^3^ = 0.480
Mixed feeding No Yes	142 (35) 261 (64)	114 (41) 163 (59)	15 (21) 55 (79)	13 (23) 43 (77)	P^0^ = 0.001 *; P^1^ = 0.002 * P^2^ = 0.012 *; P^3^ = 0.811

*n* stands for number of participants; CF stands for complementary feeding; HCW stands for healthcare workers; *n* stands for number of participants; *p* stands for probability level; G1, G2, and G3 stand for Group 1, Group 2, and Group 3, respectively. P^0^ *p*-value for comparison among the 3 groups; P^1^ between G1 and G2; P^2^ between G1 and G3; P^3^ between G2 and G3; * indicates significant difference.

**Table 5 nutrients-17-01919-t005:** Unadjusted and adjusted prevalence ratios for factors associated with nutritional indicators among children.

Stunting	Unadjusted PR (95% CI)	*p*	Adjusted PR (95% CI)	*p*
Child’s age (months) <12 12–24 >24	1 (ref) 1.20 (0.92–1.57) 0.32 (0.16–0.65)	0.178 0.002 *	1 (ref) 1.18 (0.90–1.55) 0.33 (0.16–0.65)	0.239 0.001 *
Child’s sex Girls Boys	1 (ref) 1.24 (0.97–1.58)	0.083	1 (ref) 1.27 (1.00–1.61)	0.049 *
DDS <4 ≥4	1 (ref) 0.71 (0.51–0.99)	0.041 *	1 (ref) 0.72 (0.52–0.99)	0.041 *
House type Non-brick Brick	1 (ref) 1.14 (0.90–1.45)	0.227	1 (ref) 1.24 (0.98–1.57)	0.069
**Underweight**	**Unadjusted PR (95% CI)**	** *p* **	**Adjusted (95%CI)**	** *p* **
Child’s sex Girls Boys	1 (ref) 1.96 (1.59–2.42)	≤0.0001 *	1 (ref) 2.40 (1.40–4.11)	0.001 *
DDS <4 ≥4	1 (ref) 0.47 (0.22–1.00)	0.050 *	1 (ref) 0.43 (0.20- 0.92)	0.029 *
Household head Self-headed Spouse Partner Parents Relatives	1 (ref) 0.34 (0.15–0.79) 0.31 (0.12–0.81) 0.51 (0.26–1.00) 0.44 (0.13–1.47)	0.012 * 0.017 * 0.051 0.182	1 (ref) 0.35 (0.15–0.78) 0.33 (0.13–0.87) 0.49 (0.24–0.99) 0.43 (0.13–1.71)	0.011 * 0.024 * 0.046 * 0.254
**Thinness**	**Unadjusted PR (95% CI)**	** *p* **	**AOR (95%CI)**	** *p* **
DDS <4 ≥4	1 (ref) 3.02 (1.29–7.12)	0.011 *	1 (ref) 2.70 (1.13–6.45)	0.025 *
Child’s age <12 12–24 >24	1 (ref) 3.12 (1.17–8.28) 2.13 (0.69–6.59)	0.023 * 0.189	1 (ref) 2.80 (1.02–7.64) 1.70 (0.56–5.18)	0.045 * 0.352
**Overweight/obesity**	**Unadjusted PR (95% CI)**	** *p* **	**AOR (95%CI)**	** *p* **
Child’s age <12 12–24 >24	1 (ref) 1.89 (1.18–3.01) 0.80 (0.36–1.76)	0.008 * 0.576	1 (ref) 1.94 (1.25–3.03) 0.70 (0.32–1.70)	0.003 * 0.479
Household income <ZAR 5000 ZAR 5000–ZAR 10,000 ZAR 10,001–ZAR 15,000 >ZAR 15,000	1.24 (0.72–2.14) 1.02 (0.47–2.20) 3.24 (2.14–6.82)	0.437 0.961 ≤0.0001 *	1 (ref) 1.28 (0.74–2.20) 1.02 (0.47–2.21) 4.09 (2.33–7.17)	0.378 0.958 ≤0.0001 *

DDS stand for dietary diversity scores; ZAR is the currency code for the South African Rand; PR stands for prevalence ratio; CI stands for confidence interval; * indicates significant association (*p* < 0.05).

## Data Availability

The dataset for the study group generated and analyzed during the current study is available from the corresponding author upon reasonable request due to ethical restrictions.

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
