# Peer review of "Nutritional Challenges Among Children Under Five in Limpopo Province, South Africa: Complementary Feeding Practices and Dietary Diversity Deficits"

_nutrients, 2025, doi:10.3390/nu17111919_

Round 1
Reviewer 1 Report
Comments and Suggestions for Authors
This is a cross-sectional study with adequate novelty, which assessed complementary feeding practices and dietary diversity in relation to the nutritional status of children under five attending health facilities. This study is well-organized and deals with an interesting topic. It is well-written and just minor revisions should be performed.
In the limitations of the study, the authors should report that there are several other confounding factors that they could be included in the multivariate analysis and which were not available in this study. In addition, the authors should report as a limitation the fact that they used self-reported data (even via interviews) which may lead to recall bias. Moreover, minor English language editing is recommended.
Author Response
REVIEWER 1
This is a cross-sectional study with adequate novelty, which assessed complementary feeding practices and dietary diversity in relation to the nutritional status of children under five attending health facilities. This study is well-organized and deals with an interesting topic. It is well-written and just minor revisions should be performed. In the limitations of the study, the authors should report that there are several other confounding factors that they could be included in the multivariate analysis and which were not available in this study. In addition, the authors should report as a limitation the fact that they used self-reported data (even via interviews) which may lead to recall bias. Moreover, minor English language editing is recommended.
Response: We appreciate the reviewer’s insightful comments. We have revised the limitations section to include the points raised, specifically noting the potential influence of unmeasured confounding factors (lines 459 – 461) and the limitations associated with self-reported data, including the possibility of recall bias (lines 455-457). Additionally, we have had the manuscript reviewed by a native English-speaking colleague and further proofread it using Copilot to enhance clarity and readability.
Reviewer 2 Report
Comments and Suggestions for Authors
The manuscript entitled “ nutrients-3672111_Nutritional Challenges among Children Under Five in Limpopo Province, South Africa: Complementary Feeding Practices and Dietary Diversity Deficits” is submitted for publication in the section “Nutrition in Women ” of the special issue “ Perinatal Outcomes and Early-Life Nutrition” of the journal “ Nutrients”.
The study addresses a critical issue: despite numerous interventions, significant nutritional challenges persist among children in rural South Africa. This cross-sectional study, conducted in Thabazimbi (n = 409), investigated complementary feeding practices and dietary diversity in children under five years of age. The findings revealed that 38% of the children were stunted, 13% were underweight, and 17% were overweight. Only 27% had been exclusively breastfed, and 76% did not meet the minimum dietary diversity requirements. These results underscore the role of inadequate feeding practices in poor nutritional outcomes, highlighting the urgent need for targeted and context-specific nutrition interventions.
The content of this manuscript aligns with the focus and scope of the selected section and special issue.
Reviewer Comments:
Abstract:
Each section of the abstract must contain specific and clearly delineated content. The objective should not include methodological elements, and the results should be distinct from the methodology. The abstract must be rewritten accordingly, ensuring that each part—objective, methods, results, and conclusions—addresses its specific purpose. Additionally, the objective must be stated with precision and clarity.
Keywords:
Please revise the keywords according to the Medical Subject Headings (MeSH) classification.
Introduction:
The introduction presents the importance and context of the topic well and uses relevant literature. However, the transition to the study’s hypothesis and objective is vague. This final portion of the introduction should be rewritten to clearly and precisely define the hypothesis and the specific objectives of the study.
Materials and Methods:
This section includes several comments that belong in either the discussion or the introduction. For instance, the statement in line 88 that this was originally a master’s thesis is not necessary for a journal publication and should be omitted. The section should begin with the study design, followed by the conceptual framework. Thus, lines 88–94 should be rewritten accordingly.
Lines 95–104 describe methodological procedures but do not belong under study design or conceptual framework.
In subsection 2.2, there is insufficient detail about data collection methods—whether it was conducted in schools, at the household level, through direct interviews, or self-administered questionnaires.
Lines 138–139 mention that participating children had mothers attending clinics. Does this imply the mothers were ill? Clarification is needed.
Line 142 refers to random sampling—please describe the randomization process in detail.
In line 162, the use of an adapted questionnaire is mentioned. Please provide the reference for this instrument. Information collected from mothers should be clearly separated from the anthropometric assessment of children. Describe in detail how children's anthropometric measurements were taken. This information should not be duplicated in the subsequent section (line 253 onwards).
Statistical Analysis:
Given that the study uses a cross-sectional design, prevalence ratios are more appropriate than odds ratios. Please revise accordingly.
Results:
Tables 1, 3, and 4 should include comparative analyses among different categories of the studied variables to determine whether significant differences exist. In Table 6, unadjusted prevalence ratios should be presented before adjusted ones. All tables and figures must include full definitions of any abbreviations in footnotes.
Discussion:
The discussion is well-written and appropriately addresses the study’s limitations.
Conclusion:
The conclusion should be revised to directly reflect the answer to the study objective.
General Comments:
This is an interesting and valuable study that provides important insights into dietary diversity among children under five in a specific region. However, several sections contain information that is either misplaced or missing. It is recommended that the authors revise the manuscript thoroughly to ensure that each section includes only relevant and complete information.
Author Response
Reviewer Comments:
Abstract:
Each section of the abstract must contain specific and clearly delineated content. The objective should not include methodological elements, and the results should be distinct from the methodology. The abstract must be rewritten accordingly, ensuring that each part—objective, methods, results, and conclusions—addresses its specific purpose. Additionally, the objective must be stated with precision and clarity.
Response: We thank the reviewer for the valuable feedback regarding the structure and clarity of the abstract. In response, we have thoroughly revised the abstract to ensure that each section, objective, methods, results, and conclusions, addresses its specific purpose clearly and distinctly. Specifically, we have removed methodological elements from the objective, clarified the objective statement for precision, and ensured that the results are presented separately from the methods. These revisions have been implemented in lines 6 - 32 of the revised manuscript.
Keywords:
Please revise the keywords according to the Medical Subject Headings (MeSH) classification.
Response: The keywords have been revised to align with the Medical Subject Headings (MeSH) classification. The updated list includes terms that accurately reflect the core concepts of the study and enhance discoverability in biomedical databases.
Materials and Methods:
This section includes several comments that belong in either the discussion or the introduction. For instance, the statement in line 88 that this was originally a master’s thesis is not necessary for a journal publication and should be omitted. The section should begin with the study design, followed by the conceptual framework. Thus, lines 88–94 should be rewritten accordingly.
Response: The section has been rewritten and is in lines 91 - 106.
Lines 95–104 describe methodological procedures but do not belong under study design or conceptual framework.
Response: We have shifted the lines to data collection; lines 181 - 183.
In subsection 2.2, there is insufficient detail about data collection methods—whether it was conducted in schools, at the household level, through direct interviews, or self-administered questionnaires.
Response: The study was conducted in health facilities, indicated throughout the methods and material sections in lines 85; 93; 127; 144; 150; 152; 195.
Lines 138–139 mention that participating children had mothers attending clinics. Does this imply the mothers were ill? Clarification is needed.
Response: Clarified in lines 128 -129, and 167 – 170.
Line 142 refers to random sampling—please describe the randomization process in detail.
Response: described in lines 149 – 160: A multistage sampling strategy was employed to ensure representativeness across the sub-district. In the first stage, health facilities were selected using stratified random sampling, considering geographic distribution, patient volume, and accessibility. A list of all public health facilities in the sub-district was stratified by location, and facilities were randomly selected from each stratum using a random number generator to ensure geographic and service-level diversity. In the second stage, systematic random sampling was used to recruit eligible participants within each facility. Upon arrival at each clinic, a sampling interval was determined based on the average daily number of eligible mothers attending child health services. Every third biological mother in the clinic waiting area was approached after completing her consultation and invited to participate in the study. The starting point was randomly selected each day by drawing a number between 1 and 3.
In line 162, the use of an adapted questionnaire is mentioned. Please provide the reference for this instrument. Information collected from mothers should be clearly separated from the anthropometric assessment of children. Describe in detail how children's anthropometric measurements were taken. This information should not be duplicated in the subsequent section (line 253 onwards).
Response: Socio-demographic data were collected using a structured questionnaire adapted from previously validated instruments used in maternal and child nutrition research (References of studies adapted from are included in lines 207-208). We have also expanded the description of the anthropometric measurement procedures to provide greater detail, as requested lines 244 – 263. We separated the mothers’ section from that of children on anthropometric measurements in lines 265-288.
Statistical Analysis:
Given that the study uses a cross-sectional design, prevalence ratios are more appropriate than odds ratios. Please revise accordingly.
Response: We have revised the statistical analysis section, and added relevant information on further statistical comparison and prevalence ratios; lines 299 – 311. Fisher’s exact test when expected cell counts were below five. Socioeconomic and feeding practice variables were compared across child age groups, and nutritional indicators were examined by sex and age group. To explore associations between nutritional outcomes and explanatory variables, univariate and multivariate Poisson regression models with robust standard errors were used to estimate crude (PR) and adjusted prevalence ratios (aPR) with 95% confidence intervals (CIs). Variables with p-values ≤ 0.25 in univariate analysis were included in the multivariate models using a stepwise backward elimination approach. Potential confounders at the child, maternal, and household levels were adjusted for in the final models to improve the accuracy of the estimates. Results are presented as medians with interquartile ranges (IQRs) for continuous variables, frequencies and percentages for categorical variables, PR and aPR with 95% CIs for regression models. Statistical significance was set at p < 0.05.
Results:
Tables 1, 3, and 4 should include comparative analyses among different categories of the studied variables to determine whether significant differences exist.
Response: We have revised Tables 1 (lines 312 – 333); 3 (lines 347-257) and 4 (lines 369 – 372) and added comparisons among and between groups and different categories with p-values. As a result, the narratives for Tables of concern were also revised. Footnotes have been added under tables.
In Table 6, unadjusted prevalence ratios should be presented before adjusted ones. All tables and figures must include full definitions of any abbreviations in footnotes.
Response: We have revised the statistical analysis of the paper, including using the Poisson regression to generate unadjusted and adjusted PR presented in table 6 with a rewritten narrative; lines 393 - 408
Discussion:
The discussion is well-written and appropriately addresses the study’s limitations.
Response: Thank you. But due to the revised results, we also revise the discussion to align with comparison and prevalence ratios we reported; lines 409 - 477. Limitations were slightly expanded as per the request of the other reviewer; lines 479 – 481 and 483 - 485.
Conclusion:
The conclusion should be revised to directly reflect the answer to the study objective.
Response: We have revised the conclusion in lines 495 – 507.
General Comments:
This is an interesting and valuable study that provides important insights into dietary diversity among children under five in a specific region. However, several sections contain information that is either misplaced or missing. It is recommended that the authors revise the manuscript thoroughly to ensure that each section includes only relevant and complete information.
Round 2
Reviewer 1 Report
Comments and Suggestions for Authors
The authors have sufficiently improved their manuscript.
Author Response
Reviewer 1
Thank you
Reviewer 2 Report
Comments and Suggestions for Authors
I have carefully reviewed the revised version of the manuscript as well as the authors’ responses to the suggestions provided.
With regard to the Materials and Methods section, the content from lines 94 to 98 does not correspond to methodological details and should therefore be relocated to the Introduction. Similarly, the content from lines 103 to 106 would be more appropriately placed in the Discussion section.
Concerning the data collection procedure involving children, the authors repeatedly state: "The study was conducted in health facilities, indicated throughout the methods and material sections." However, this does not clarify whether the children included in the study were attending the health facilities due to health problems. If that were the case, it could introduce a selection bias and potentially overestimate the prevalence of inadequate complementary feeding practices and dietary diversity deficits. The authors must clearly specify the characteristics of the child population sampled for this study, including the settings and conditions under which the data were collected, as this is a critical aspect in the interpretation of the results.
In the Statistical Analysis section, the authors should indicate which test was used to assess the normality of the data.
It is also worth noting that Figure 1 appears to be identical to figures presented in other publications, which raises concerns about originality and potential duplication.
Despite these issues, I consider that the authors have substantially improved the overall quality of the manuscript.
Author Response
With regard to the Materials and Methods section, the content from lines 94 to 98 does not correspond to methodological details and should therefore be relocated to the Introduction. Similarly, the content from lines 103 to 106 would be more appropriately placed in the Discussion section.
Response: The content from lines 94–98 has been relocated to the Introduction section (now lines 49–58) to improve the logical flow of the manuscript. Similarly, the content from lines 104–105 has been moved to the Discussion section (now lines 442–444), rephrased and where it is more appropriately contextualized.
Concerning the data collection procedure involving children, the authors repeatedly state: "The study was conducted in health facilities, indicated throughout the methods and material sections." However, this does not clarify whether the children included in the study were attending the health facilities due to health problems. If that were the case, it could introduce a selection bias and potentially overestimate the prevalence of inadequate complementary feeding practices and dietary diversity deficits. The authors must clearly specify the characteristics of the child population sampled for this study, including the settings and conditions under which the data were collected, as this is a critical aspect in the interpretation of the results.
Response: We appreciate the reviewer’s important observation. To clarify, the study was conducted among children under five years of age attending primary health care facilities for routine child health services, specifically for immunization and growth monitoring, as stated in lines 104-105, 127-128, 134-135, 145–146, and 169–170. These services are part of South Africa’s standard preventive child health care and are not related to illness. Furthermore, as outlined in the exclusion criteria, children with known chronic illnesses or congenital conditions affecting growth or feeding were excluded from participation. Therefore, the study sample comprised presumably healthy children, and the concern regarding selection bias due to the inclusion of children with health problems does not apply in this context.
In the Statistical Analysis section, the authors should indicate which test was used to assess the normality of the data.
Response: As indicated in the Statistical Analysis section (lines 301–302), the Shapiro–Francia test was used to assess the normality of continuous variables. We have ensured that this information is clearly stated for transparency and reproducibility.
It is also worth noting that Figure 1 appears to be identical to figures presented in other publications, which raises concerns about originality and potential duplication.
Response: We appreciate the reviewer’s feedback and acknowledge the concern regarding originality. Our intention with Figure 1 which is our original art, was to clearly illustrate the step-by-step recruitment process using a visual format that is easy to follow. While we used a SmartArt graphic for clarity, we did not anticipate that its structure might resemble figures in other studies, as the content is specific to our research. Nonetheless, we have revised the figure to present the information in a more distinct format.
Despite these issues, I consider that the authors have substantially improved the overall quality of the manuscript.